# The Diverse Binding Modes Explain the Nanomolar Levels of Inhibitory Activities Against 1-Deoxy-d-Xylulose 5-Phosphate Reductoisomerase from *Plasmodium falciparum* Exhibited by Reverse Hydroxamate Analogs of Fosmidomycin with Varying *N*-Substituents

**DOI:** 10.3390/molecules30010072

**Published:** 2024-12-28

**Authors:** Sana Takada, Mona A. Abdullaziz, Stefan Höfmann, Talea Knak, Shin-ichiro Ozawa, Yasumitsu Sakamoto, Thomas Kurz, Nobutada Tanaka

**Affiliations:** 1School of Pharmacy, Kitasato University, Minato-ku, Tokyo 108-8641, Japanozawas@pharm.kitasato-u.ac.jp (S.-i.O.); 2Institute of Pharmaceutical and Medicinal Chemistry, Faculty of Mathematics and Natural Sciences, Heinrich Heine University Düsseldorf, Universitätsstr. 1, 40225 Düsseldorf, Germany; 3National Research Centre (NRC), 33 El Buhouth St., Ad Doqi, Dokki, Cairo 12622, Egypt; 4School of Pharmacy, Iwate Medical University, Yahaba, Morioka 028-3694, Japan; sakamoto@stbio.org

**Keywords:** crystal structure, DXR, fosmidomycin, IspC, malaria

## Abstract

It is established that reverse hydroxamate analogs of fosmidomycin inhibit the growth of *Plasmodium falciparum* by inhibiting 1-deoxy-d-xylulose 5-phosphate reductoisomerase (DXR), the second enzyme of the non-mevalonate pathway, which is absent in humans. Recent biochemical studies have demonstrated that novel reverse fosmidomycin analogs with phenylalkyl substituents at the hydroxamate nitrogen exhibit inhibitory activities against *Pf*DXR at the nanomolar level. Moreover, crystallographic analyses have revealed that the phenyl moiety of the *N*-phenylpropyl substituent is accommodated in a previously unidentified subpocket within the active site of *Pf*DXR. In this study, the crystal structures of *Pf*DXR in complex with a series of reverse *N*-phenylalkyl derivatives of fosmidomycin were determined to ascertain whether the high inhibitory activities of the derivatives are consistently attributable to the utilization of the subpocket of *Pf*DXR. While all reverse fosmidomycin derivatives with an *N*-substituted phenylalkyl group exhibit potent inhibitory activity against *Pf*DXR, the present crystal structure analyses revealed that their binding modes to the *Pf*DXR are not uniform. In these compounds, the nanomolar inhibitory activities appear to be driven by binding modes distinct from that observed for the inhibitor containing the *N*-phenylpropyl group. The structural information obtained in this study will provide a basis for further design of fosmidomycin derivatives.

## 1. Introduction

Malaria is one of the world’s three major infectious diseases, caused by *Plasmodium* species transmitted to humans through the bites of female *Anopheles mosquitoes* [1]. The five species of *Plasmodium* known to infect humans are *P. falciparum*, *P. vivax*, *P. ovale*, *P. malariae*, and *P. knowlesi*. Of these, *P. falciparum* malaria is most prevalent in sub-Saharan Africa and is associated with the most severe symptoms [1]. As indicated in the WHO World Malaria Report 2023, approximately 608,000 individuals perish annually from malaria, with approximately 95% of these fatalities occurring in Africa. The emergence of multidrug-resistant *P. falciparum*, including strains resistant to artemisinin, the current first-line treatment, highlights the urgent necessity for the development of novel antimalarial drugs with alternative mechanisms of action [1].

1-Deoxy-d-xylulose 5-phosphate reductoisomerase from *Plasmodium falciparum* (*Pf*DXR or *Pf*IspC) is a homodimeric enzyme, with each subunit consisting of 488 amino acid residues. The enzyme DXR is also known as IspC, derived from the gene name *ispC* in the isoprenoid biosynthesis pathway. It is the enzyme responsible for the second step of a six-step catalytic reaction sequence catalyzing the conversion of the substrate 1-deoxy-d-xylulose 5-phosphate (DOXP) to 2-*C*-methyl-d-erythritol 4-phosphate (MEP) in the non-mevalonate pathway (MEP pathway) [2]. In the MEP pathway, which is found in pathogens such as *Plasmodium* spp. and *Mycobacterium tuberculosis*, isopentenyl diphosphate (IPP) and dimethylallyl diphosphate (DMAPP), which are precursors for isoprenoid biosynthesis, are synthesized starting from pyruvate and glyceraldehyde-3-phosphate via the intermediate MEP [3]. In contrast, mammals employ the mevalonate (MVA) pathway to generate IPP and DMAPP. Accordingly, DXR, which belongs to the MEP pathway, a metabolic pathway not present in mammals, is regarded as a promising target for drug discovery.

Fosmidomycin is the first and most well-studied inhibitor of DXR. The antibiotic properties of this compound were first documented in 1980 [4]. Subsequently, Kuzuyama et al. [5] and Zeidler et al. [6] demonstrated that fosmidomycin is a DXR inhibitor. In 1999, Jomaa et al. reported that fosmidomycin and FR900098 (an *N*-acetyl derivative of fosmidomycin) suppress the growth of *P. falciparum* by inhibiting *Pf*DXR [7]. Since then, among the enzymes of the MEP pathway, DXR has been the subject of particular attention as a potential molecular target for the development of novel antimalarial drugs [8].

*Pf*DXR is localized in the apicoplast, and inhibition of this protein in the asexual blood stage of *P. falciparum* has been demonstrated to be lethal for the parasite [9]. The *Pf*DXR subunit is comprised of three domains and a linker region: an NADPH-binding domain (residues 77-230), a catalytic domain (231-369), a C-terminal domain (396-488), and a linker region (370-395) [10]. For the catalytic function of DXR, one molecule of NADPH and one divalent metal ion (either Mn^2+^ or Mg^2+^) are required per subunit as cofactors. The NADPH-binding domain is comprised of a typical nucleotide-binding motif, the Rossmann fold. To date, 77 DXR structures from 10 species, including *P. falciparum*, *Mycobacterium tuberculosis*, and *Escherichia coli*, have been deposited in the Protein Data Bank (PDB). Of these, 21 structures are derived from *P. falciparum*, and they have been reported by various research groups between 2011 and 2024. *Pf*DXR shares approximately 50% sequence similarity with DXR of other species, and the overall fold is similar among the DXRs.

The first crystal structures of DXR were determined for *E. coli* DXR (*Ec*DXR) by two independent research groups in 2002 [11,12]. Subsequently, the structure of DXR from *M. tuberculosis* (*Mt*DXR) was elucidated in 2007 [13]. Following these, the crystal structures of the ternary (enzyme–metal–NADPH) and quaternary (enzyme–metal–NADPH-inhibitor) complexes of *Pf*DXR were finally determined in 2011 [14], thereby elucidating the molecular mechanism by which fosmidomycin inhibits *P. falciparum* growth. The crystal structure of *Pf*DXR in complex with fosmidomycin revealed that the binding of the inhibitor caused the enzyme molecule to assume a closed conformation, thereby fixing the flexible loop (residues 291 to 299) of the active site. The indole ring of Trp296 in the flexible loop formed strong van der Waals (vdW) contacts with the alkyl moiety of fosmidomycin. Furthermore, the crystal structure of *Pf*DXR in complex with FR900098, the initial structure of a DXR in complex with FR900098, was also determined. In this complex, the flexible loop of *Pf*DXR also exhibited a closed conformation, as observed in the fosmidomycin complex [14]. In contrast, the crystal structure of *Pf*DXR in complex with an α-phenyl fosmidomycin derivative with a reverse hydroxamate revealed that the bulky α-phenyl moiety induced a relatively open conformation of the flexible loop [15]. Furthermore, the open and disordered conformations of the flexible loop were observed for *Pf*DXRs in complex with other α-substituted [16,17] and β-substituted [18,19] fosmidomycin derivatives. Similarly, disorders of flexible loops have been observed in the context of α-substituted fosmidomycin derivatives in complex with DXR from other species, including *M. tuberculosis* [20,21,22] and *E. coli* [23]. The open conformation of the flexible loop, in which Trp296 was flipped out, created space around the hydroxamate nitrogen, which was not observed in the fosmidomycin complex [14,15]. Based on this observation, a new series of inhibitors was synthesized, in which a phenylalkyl group was introduced into the hydroxamate nitrogen, with the objective of exploiting this space. The abilities of these inhibitors to inhibit the *Pf*DXR enzymatic activity and *P. falciparum* growth were then evaluated [24]. Moreover, for compound **13e** (referred to henceforth as MAMK89 in this work, as it was in our recent publication [24]), which exhibited the most potent antiplasmodial activity of these compounds, we conducted crystal structure analyses in complex with *Pf*DXR. The analyses suggested that the potent activity of MAMK89, as compared with that of fosmidomycin, is enhanced by the interaction with a subpocket that was not utilized by the previously reported fosmidomycin derivatives. In the active site of *Pf*DXR, the phenylpropyl group of MAMK89 assumed a bent conformation, with the phenyl moiety entering the subpocket and establishing vdW contacts with the side chains of Ser232, Pro358, and Pro363. Additionally, the phenyl ring engaged in a CH-π interaction with the imidazole ring of His341.

In this study, the objective is to elucidate the molecular mechanism underlying the structure–activity relationship of the *N*-substituted phenylalkyl derivatives of fosmidomycin, as reported in our recent work [24]. To this end, the crystal structures of various MAMK89 analogs in complex with *Pf*DXR were elucidated. The MAMK89 analogs with the following structural modifications were subjected to crystal structure analyses in complex with *Pf*DXR: (i) alterations in the linker length (C2, C4, and C5) of the *N*-phenylalkyl group and (ii) introduction of substituents to the terminal phenyl moiety (*para* methyl and *para* methoxy) of the *N*-phenylpropyl group (Figure 1). The high-resolution crystal structure determination was successfully achieved for all complexes, with a minimum resolution of 1.84 Å and a maximum resolution of 1.33 Å. The binding free energy for each inhibitor to the active site of *Pf*DXR was also estimated using the reliable crystal structures for the series of complexes.

## 2. Results

### 2.1. Crystal Structure Analyses

The cocrystallization of the ternary (enzyme–Mn^2+^–inhibitor) and quaternary (enzyme–Mn^2+^–NADPH–inhibitor) complexes of *Pf*DXR yielded crystals with a space group of *P*2_1_ for all complexes. The crystals diffracted to resolutions ranging from 1.33 to 1.84 Å (Appendix A). The initial phase determination and structure refinement were conducted for the MAMK150–quaternary complex of *Pf*DXR using a difference Fourier method with the coordinate set of the re-refined (see below) MAMK89–quaternary complex of *Pf*DXR (Figure 2A) as the initial model (PDB ID: 9JOB). The refined MAMK150–quaternary complex was then employed as a template for the refinement of the structures of the remaining complexes. The asymmetric units of the crystals of all complexes comprised a *Pf*DXR dimer. The two subunits (A and B) of the *Pf*DXR dimer exhibited equivalent structures, and the electron densities of the bound ligand molecules were observed in the active site of both subunits of all complexes. As the average temperature factor of the atoms belonging to subunit B was lower than that of subunit A, the following description will primarily focus on subunit B of each complex. The interactions between the hydroxamate and phosphonate moieties of a series of inhibitors used in this study and the active site of *Pf*DXR were analogous to those observed for several fosmidomycin analogs in complex with *Pf*DXR. With the aim of providing a representative example, the interactions between the hydroxamate and phosphonate moieties of MAMK150 and the active site of the subunit B of *Pf*DXR are described below. The oxygen atoms of the hydroxamate group of MAMK150 coordinated an Mn^2+^ ion, which was bound by the side chains of Asp231, Glu233, and Glu315. The Mn^2+^ ion in the MAMK150–quaternary complex exhibited a trigonal bipyramidal geometry, with three amino acid side chains and two inhibitor atoms at distances of 2.1 to 2.2 Å. The phosphonate group formed a hydrogen-bond network with the side chains of Ser269, Ser270, Ser306, Asn311, and Lys312, as well as the main-chain nitrogen of Ser269 and two water molecules, with distances of 2.6 to 2.9 Å. These interactions have also been extensively characterized in previous studies [13,14]. In this study, we concentrate on the interactions between the *N*-phenylalkyl substituents and the enzyme active site. The enzyme–inhibitor interactions are summarized in Appendix A.

### 2.2. MAMK89 Complexes (N-Substituent: Phenylpropyl)

The crystal structures of the MAMK89–ternary and MAMK89–quaternary complexes [24] were re-refined against the higher resolution intensity data (see Section 4) at resolutions of 1.50 Å and 1.39 Å, respectively. In consequence, the coordinate files of the ternary and quaternary complexes are updated from 8JNV and 8JNW to 9JOA and 9JOB, respectively. The overall structure of the MAMK89–quaternary complex was similar to those of previously reported quaternary complex of *Pf*DXR [14,15]. The MAMK89 molecule was located in the active site cleft of each subunit (Figure 2A). It should be noted that irrespective of the improvement in resolution, the binding mode in which the phenyl moiety of the *N*-phenylpropyl group of MAMK89 is accommodated in the subpocket of the active site of *Pf*DXR, with or without coenzyme NADPH, remained a prominent feature of the MAMK89 complex. The description in this paper pertains to subunit B of the MAMK89–quaternary complex. The phenyl ring of the phenylpropyl moiety was in vdW contact with the side chains of Ser232, Pro358, and Pro363 and exhibited a CH-π interaction with the imidazole ring of His341. In the following figures of the structures of other inhibitor complexes, the structure of the MAMK89 complex is shown for comparison (Figure 2B–F).

### 2.3. MAMK150 Complexes (N-Substituent: Phenylethyl)

The crystal structures of the MAMK150–ternary (PDB ID: 9JZ7) and MAMK150–quaternary (PDB ID: 9JZ8) complexes were determined at resolutions of 1.79 Å and 1.53 Å, respectively. The following description pertains to subunit B of the MAMK150–quaternary complex (Figure 2B). An intramolecular contact was observed between the hydroxamate *N*-phenylethyl group and the α-phenyl group of MAMK150. The ethyl linker of MAMK150 was in vdW contact with the side chains of Ser232 and Met360, which are located within the active site core. With regard to the phenyl moiety of the *N*-phenylethyl group of MAMK150, the eastern face was in vdW contact with the side chain of Met360 from the active site core and that of Met298 from the flexible loop (residues 291-299), whereas the western face was in vdW contact with the side chain of Pro358 from the active site core and that of Trp296 from the flexible loop. In addition to these interactions with the side chains of the active site residues, the edge of the phenyl ring of the *N*-phenylethyl group of MAMK150 was in vdW contact with the main-chain peptide bond plane of Pro358 to Met360. Notable differences between the MAMK150 complex and the MAMK89 complex are that the MAMK150 complex exhibited a slightly more open loop structure, and the position of the side chain of Trp296 was completely different (Figure 2B). In the MAMK150 complex, the active site subpocket of *Pf*DXR [24] was occupied by five water molecules. The *N*-phenylethyl group of MAMK150 adopted a curved conformation that did not fit into the subpocket. In contrast, in the MAMK89 complex, two water molecules at the entrance of the subpocket were displaced by the phenyl group of the *N*-phenylpropyl substituent of MAMK89, leaving three water molecules at the back. In other complexes, the subpocket was similarly occupied by five water molecules when the *N*-phenylalkyl substituent of the inhibitor adopted a curved conformation that did not bind to the subpocket.

### 2.4. MAMK218 Complexes (N-Substituent: Phenylbutyl)

The crystal structures of the MAMK218–ternary (PDB ID: 9JZ9) and MAMK218–quaternary (PDB ID: 9JZA) complexes were determined at resolutions of 1.33 Å and 1.55 Å, respectively. In comparison to the MAMK150–quaternary complex, the MAMK218 complexes exhibited increased flexibility. The residues 293-295 and 293-297, respectively, in the flexible loops of the ternary and quaternary complexes were observed to be disordered and, therefore, not modeled. The following description pertains to subunit B of the MAMK218–ternary complex (Figure 2C). The most notable feature of the MAMK218 complex was the observation of two distinct conformers for the *N*-phenylbutyl group of MAMK218, namely a bent and a curved conformer. Additionally, Trp296 exhibited two conformers, corresponding to the two conformers observed for MAMK218. The phenyl moiety of the *N*-phenylbutyl group was accommodated in the subpocket in the case of the bent conformer, whereas in the case of the curved conformer, it was located in the center of the active site cleft. The intermolecular interactions observed for the phenyl moiety of the *N*-phenylbutyl group in the bent conformer were found to be equivalent to those observed for the phenyl moiety of the *N*-phenylpropyl group in the MAMK89 complexes [24]. The side chain of Trp296 involved in the recognition of the bent conformer of MAMK218 exhibited a similar conformation to that in the MAMK89 complex (Figure 2C). The phenyl ring was in contact with the subpocket through vdW interactions with the side chains of Ser232, Pro358, and Pro363. Additionally, the phenyl ring was engaged in a CH-π interaction (T-shape stacking) with the imidazole ring of His341. The interactions between the curved conformer of MAMK218 and *Pf*DXR were analogous to those observed for the MAMK150–quaternary complex. In the case of MAMK218, the phenyl moiety of the *N*-phenylbutyl substituent was somewhat exposed to the protein surface, attributed to the two-carbon extension of the linker length of the *N*-substituent relative to MAMK150. Additionally, the terminal end of the *N*-phenylbutyl substituent in MAMK218 displayed a degree of flexibility. The temperature factor of the phenyl moiety of the *N*-phenylbutyl substituent in MAMK218 was higher than that of the core part of MAMK218, and the electron density of the phenyl moiety was found to be relatively weak.

### 2.5. MAMK251 Complexes (N-Substituent: Phenylpentyl)

The crystal structures of the MAMK251–ternary (PDB ID: 9JZB) and MAMK251–quaternary (PDB ID: 9JZC) complexes were determined at resolutions of 1.47 Å and 1.77 Å, respectively. The residues 293-296 of the subunit A of the ternary complex and the residues 293-295 of both subunits of the quaternary complex were found to be disordered and, therefore, not modeled. The following description pertains to subunit B of the MAMK251–ternary complex (Figure 2D). A notable feature of the MAMK251 complex is the observation of stable binding of the *N*-phenylpentyl substituent despite the alkyl linker moiety being two methylene groups longer than that of MAMK218. An intramolecular contact was observed between the *N*-pentyl linker moiety and the α-phenyl group of MAMK251. The pentyl linker of MAMK251 was in vdW contact with the side chains of Ser232, Pro358, and Met360, which are located in the active site core, as well as with those of Trp296 and Met298, which are located in the flexible loop. The phenyl moiety of the *N*-phenylpentyl group of MAMK251 was in vdW contact with the side chains of Ser88, Asn92, and Trp296. Additionally, a vdW contact was observed between the benzyl moiety of the *N*-phenylpentyl group of MAMK251 and the main-chain peptide bond plane of Pro358 to Met360. In contrast to the MAMK218 complex, only the curved conformer of the inhibitor was observed in the case of the MAMK251 complex. The conformation of Trp296 in the MAMK251 complex was completely different from that in the MAMK89 complex (Figure 2D) and similar to that in the MAMK150 complex.

### 2.6. MAMK433 Complex (N-Substituent: (p-Tolyl)propyl)

The crystal structure of the MAMK433–quaternary (PDB ID: 9JZD) complex was determined at a resolution of 1.65 Å. The residues 293-295 of both subunits of the MAMK433–quaternary complex were found to be disordered and, therefore, not modeled. The following description pertains to subunit B of the MAMK433–quaternary complex (Figure 2E). It is notable that the *p*-tolyl group was not accommodated in the subpocket despite the propyl linker of the *N*-substituent being equal to that of MAMK89. The enzyme–inhibitor interactions observed in the MAMK433–quaternary complex were similar to those observed in the MAMK150–quaternary complex. An intramolecular contact was observed between the *N*-(*p*-tolyl)propyl group and the α-phenyl group of MAMK433. The propyl linker of MAMK433 was in vdW contact with the side chains of Ser232 and Met360, which are located within the active site core. With regard to the *p*-tolyl moiety of the *N*-(*p*-tolyl)propyl group of MAMK433, the eastern face was in vdW contact with the side chains of Met360 and Met298, whereas the western face was in vdW interactions with the side chains of Pro358 and that of Trp296. The residues 293-295 were observed to be disordered in the MAMK433 complex. However, the position of the Trp296 side chain was found to be similar to that observed in the MAMK150 complex (Figure 2B,E).

### 2.7. MAMK431 Complex (N-Substituent: (4-Methoxyphenyl)propyl)

The crystal structure of the MAMK431–quaternary (PDB ID: 9JZE) complex was determined at a resolution of 1.84 Å. The residues 293-296 of both subunits of the MAMK431-quaternary complex were found to be disordered and, therefore, not modeled. The following description pertains to subunit B of the MAMK431–quaternary complex (Figure 2F). The enzyme–inhibitor interactions observed in the MAMK431–quaternary complex were comparable to those observed in the MAMK433–quaternary complex. Similarly to the MAMK433 complex, the 4-methoxyphenyl group was unable to be accommodated in the subpocket despite the linker length of the *N*-substituent being equal to that of MAMK89.

### 2.8. The Binding Free Energies of Inhibitor Molecules

To elucidate the structure–activity relationships of the compounds under investigation, we conducted a binding free energy analysis of the binding of each compound to the active site of *Pf*DXR. The binding free energies (Δ*G*_bind_) for the inhibitor molecules were calculated by the MM-GBSA method [25,26] using the crystal structures obtained in this study as template models. This approach is a rapid force-field-based technique that uses a generalized Born and surface area continuum (implicit) solvation solvent model to determine the Δ*G*_bind_ from the difference between the energies of the complex, ligand, and protein in solution. It is an efficient end-point free energy method that ignores details of the binding pathway and estimates the free energy based only on the bound and unbound states. The binding analyses demonstrated that the binding free energies of all the compounds were comparable, with an approximate value of −40 kcal/mol (Table 1). These values are in agreement with those previously reported for the binding of lipophilic fosmidomycin derivatives with non-hydroxamate metal-binding groups to *Ec*DXR, which are approximately minus tens of kcal/mol [27]. As a logical consequence, comparable binding free energies are estimated when ligands of analogous sizes bind to analogous active sites despite the differences in enzyme species and ligand basic skeleton. It should be added here that the binding energies of the MAMK218–quaternary complex (9JZA) and the MAMK431–quaternary complex (9JZE) were estimated to be within the range of −18 to −33 kcal/mol, which was similar to the binding energies of fosmidomycin (−14 to −17 kcal/mol) but with significantly different IC_50_ values (Table 1). This was due to the fact that in the case of the MAMK218–quaternary complex (9JZA) and the MAMK431–quaternary complex (9JZE), the coordinates of Trp296 in the flexible loop were absent due to disorder, thereby preventing the proper estimation of the binding energy through calculation. In other words, the binding energies were underestimated in these complexes.

## 3. Discussion

In our previous study [24], we determined the crystal structures of the MAMK89–ternary and MAMK89–quaternary complexes of *Pf*DXR and proposed that the potent activity of MAMK89 as compared with that of fosmidomycin is enhanced by the interaction with a subpocket that was not utilized by the previously reported fosmidomycin derivatives. In the active site of *Pf*DXR, the *N*-phenylpropyl substituent of MAMK89 exhibited a bent conformation, with the phenyl moiety entering the subpocket and establishing vdW contacts with the side chains of Ser232, Pro358, and Pro363. Additionally, the phenyl ring engaged in a CH-π interaction with the imidazole ring of His341.

The primary objective of this study is to ascertain the optimal length of the alkyl chain of the *N*-phenylalkyl substituent necessary to access the subpocket (Figure 1). To clarify this matter, the crystal structures of MAMK150, MAMK218, and MAMK251 in complex with *Pf*DXR were determined and compared to that of MAMK89 in complex with *Pf*DXR (Figure 3A,C). The interactions between the hydroxamate and phosphonate moieties of the series of inhibitors used in this study and the active site of *Pf*DXR are analogous (Appendix A). It is important to note, however, that the *N*-phenylalkyl group binds to the active site of *Pf*DXR in different conformations depending on the length of the alkyl chain, with the ethyl (MAMK150), propyl (MAMK89), butyl (MAMK218), and pentyl (MAMK251) groups exhibiting distinct conformations. A comparison of the four inhibitor complexes revealed that the MAMK89 complex exhibited exclusively a bent conformation, with the phenyl moiety of the *N*-phenylpropyl substituent accommodated in the subpocket (Figure 2 and Figure 3). This suggests that the length of the propyl linker is optimal for the phenyl moiety of the *N*-phenylalkyl substituent to utilize the subpocket. In the case of the MAMK218 complex, the *N*-phenylbutyl substituent exhibited two distinct conformations, with approximately half of the inhibitor molecules utilizing the subpocket. The butyl linker length in MAMK218 appears to be slightly longer than necessary to effectively utilize the subpocket. In the case of the MAMK150 and MAMK251 complexes, the linker lengths are insufficiently long (ethyl) or excessively long (pentyl), respectively, and the phenyl moiety of the *N*-substituents is unable to utilize the subpocket. While the linker length can explain whether or not the phenyl moiety of the *N*-substituent can utilize the subpocket, the correlation between the inhibitory activities and linker lengths of the inhibitors cannot be explained simply by whether or not the subpocket is utilized alone.

In a previous study, we proposed that the high inhibitory activity of a series of compounds was derived from the utilization of the subpocket. Our crystal structure analyses of the MAMK89–ternary and MAMK89–quaternary complexes corroborated this hypothesis, confirming the binding of the phenyl moiety of the *N*-phenylpropyl group into the subpocket without or with the cofactor present [24]. However, in the case of MAMK218 (35 nM) and MAMK251 (29 nM), which have inhibitory activities (IC_50_ against *Pf*DXR) comparable to that of MAMK89 (30 nM) (Figure 1), this study revealed that their phenyl groups do not bind preferentially to the subpocket or not at all (Figure 2). It can be postulated that the cumulative interaction between the *N*-substituent and the active site residues is a determining factor in the inhibitory activity of these inhibitors. The interaction between the phenyl moiety of the *N*-phenylpropyl group and the subpocket residues (Ser232, Pro358, Pro363, and His341) in the MAMK89 complex and the interactions between the tip of the *N*-substituent and the active site residues (Ser232, Pro358, Met360, Trp296, Met298, and the main-chain peptide bond plane of Pro358 to Met360) in the MAMK150, MAMK218, and MAMK251 complexes (Figure 2B–D) contribute approximately equally. This interpretation is supported by the MM-GBSA analyses (Table 1). Previous studies have demonstrated that the introduction of bulky substituents into the α-position of fosmidomycin can significantly enhance its inhibitory activity against *Pf*DXR [16,17,23]. These observations can be attributed to the augmented intermolecular interactions within the active site.

The reason for the significantly lower inhibitory activity of MAMK150 (64 nM) in comparison to those of MAMK89 (30 nM), MAMK218 (35 nM), and MAMK251 (29 nM) (Figure 1) remains unclear based on the results of crystal structure analyses (Figure 2) and computational analyses (Table 1). With regard to the experimental observations, the side chain of Trp296 is in closer vdW contact with the side chains of Cys338 and Pro358 belonging to the active site core in the case of the MAMK150 complex when compared to the other complexes. This resulted in lower temperature factors and clearer electron densities for residues 291-299 being observed for the MAMK150–ternary and MAMK150–quaternary complexes. This indicates that the energetic disadvantage associated with the reduction in degrees of freedom on the active site loop of *Pf*DXR due to the binding of MAMK150 may be a contributing factor to the significantly lower enzyme inhibitory activity of MAMK150 against *Pf*DXR in comparison to other inhibitors.

The second objective of this study is to elucidate whether the introduction of a *para*-substituent into the phenyl moiety of the *N*-phenylpropyl group enhances the binding ability by filling the space at the rear of the subpocket with the *para*-substituent (Figure 1). To this end, the crystal structures of the MAMK433 and MAMK431 in complex with *Pf*DXR were determined and compared with those of MAMK89 in complex with *Pf*DXR (Figure 3B,D). It is noteworthy that the propyl linker is identical in all three compounds; however, the *p*-methoxyphenyl and *p*-tolyl are not located within the subpocket. The inability of the phenyl moiety of the *N*-substituent to utilize the subpockets in the MAMK433 and MAMK431 complexes suggests that the subpockets lack the capacity to accommodate the *para*-substituent introduced in the phenyl moiety (Figure 4). Nevertheless, even with MAMK89 bound, a narrow space remains at the rear of the subpocket, surrounded by hydrophilic side chains, where three water molecules are bound (Figure 4). It can thus be concluded that a less bulky substituent would be required for the inhibitor to utilize this space. The observation that MAMK433 (44 nM) and MAMK431 (24 nM) exhibit comparable *Pf*DXR inhibitory activity to that of MAMK89 (30 nM) without the use of subpockets suggests that the cumulative interaction between the *N*-substituents and the active site residues plays a role in determining the inhibitory activity (Table 1), as in the case of MAMK218 and MAMK251.

A series of crystallographic analyses conducted in this study revealed a correlation between the accommodation of the phenyl moiety of the *N*-substituted phenylalkyl group in the subpocket and the conformation of the main chain carbonyl group of Asp359. In the MAMK89 complex, the carbonyl group of Asp359 adopts an upward conformation, and the subpocket entrance is open, thereby facilitating the binding of the phenyl moiety of the *N*-phenylpropyl substituent of MAMK89 into the subpocket (Figure 5). Conversely, in the MAMK251 complex, the carbonyl group of Asp359 assumes a downward conformation, resulting in the closure of the subpocket entrance (Figure 5). The upward conformations are also partly observed for the MAMK150 (Figure 2B) and MAMK218 (Figure 2C) complexes, while the downward conformations are fully observed for the MAMK251 (Figure 2D), MAMK433 (Figure 2E), and MAMK431 (Figure 2F) complexes, and are also present to a lesser extent in the MAMK150 (Figure 2B) and MAMK218 (Figure 2C) complexes. The upward conformations of the carbonyl group of Asp359 are also observed for inhibitor-free *Pf*DXR and for fosmidomycin-bound *Pf*DXR [14], indicating that this conformation represents a major state. The downward conformation of the carbonyl group of Asp359 can be interpreted as the result of an induced fit to interact with the *N*-substituent of the inhibitor that is not accommodated in the subpocket but is bound along the active site core. The plane of the peptide bond between Asp359 and Met360 in the upward conformation is nearly parallel to the phenyl ring of the *N*-phenylpropyl substituent of MAMK89, which is bound in the subpocket. In contrast, the plane of the peptide bond in the downward conformation is nearly parallel to the pentyl linker of the *N*-phenylpentyl substituent of MAMK251, which is bound in the active site cleft (Figure 5).

In this paper, we have discussed the structure–activity relationships of the *Pf*DXR inhibitory activities of various reverse fosmidomycin derivatives with large and flexible substituents at the hydroxamic acid nitrogen. Additionally, we would like to make a few remarks on their ability to inhibit *P. falciparum* growth. The compounds of interest in this study have *Pf*DXR inhibitory activities at the nanomolar level but *P. falciparum* growth inhibitory activities at the micromolar level. Given that *Pf*DXR, the target of the inhibitors under consideration, is present in the apicoplast of *P. falciparum*, it follows that the inhibitor must traverse several membranes in order to reach the enzymes of the MEP pathway in the apicoplast. At the cellular level, it is difficult to elucidate the structure–activity relationships of diverse inhibitors exclusively based on the structural data pertaining to their complexes with the target molecule. This is because a multitude of intrinsic physical and physico-chemical properties of the compounds, including membrane permeability and compound stability, play a role. As an illustration, MAMK150 exhibits a markedly diminished capacity to inhibit *Pf*DXR among the compounds under examination in this study (Figure 1). However, with regard to its capacity to inhibit the growth of *P. falciparum*, MAMK150 exhibits efficacy that is commensurate with that of MAMK89 [24]. In contrast, MAMK218, MAMK251, MAMK431 and MAMK433 display comparable inhibitory activities against *Pf*DXR to those observed with MAMK89. However, they exhibit less than one-third of the inhibitory activity observed for MAMK89 against *P. falciparum* growth [24]. It can, therefore, be posited that the development of compounds must take into account the observed fluctuations in activity between enzyme-level and cell-level evaluations. It is postulated that the deprotonated phosphonate group is the primary cause of the low activity observed at the cellular level due to low permeability. It is hypothesized that the phosphonate groups of fosmidomycin derivatives, which are deprotonated and dianionic under physiological conditions due to their p*K*_a2_ values lower than 7.4 [10], are the primary factor contributing to the observed reduction in activity at the cellular level, potentially due to their low permeability. It may be the case that masking of the phosphonate by converting the phosphonate group into a prodrug moiety is an effective approach in this regard [10].

## 4. Materials and Methods

### 4.1. Purification

The expression and purification of *Pf*DXR were conducted in accordance with the previously described methodology [14]. *E. coli* BL21(DE3) cells harboring the pQE30 expression plasmid, which contains the DNA-encoding residues Lys75 to Ser488 of *Pf*DXR, were grown in LB media at 37 °C until an OD_600_ of 0.6 was reached. Overexpression of *Pf*DXR was induced by the addition of 0.5 mM IPTG for a period of 20 h at 20 °C. Following this, the cells were harvested by centrifugation at 8000× *g* and stored at −20 °C. The cells were then suspended in a lysis buffer (20 mM Tris-HCl, pH 8.0, 50 mM NaCl, 20 mM imidazole, 2 mM DTT, and 1 mM PMSF) and disrupted using ultrasonication on ice for 3 × 30 s. The cell extract was obtained by centrifugation at 18,000× *g* for 15 min and applied to a 1 mL HisTrap HP column (Cytiva, Marlborough, MA, USA) that had been equilibrated with buffer A (20 mM Tris-HCl pH 8.0 and 50 mM NaCl). The column was washed with 20 column volumes of buffer A. Following the washing step, *Pf*DXR was eluted with a linear gradient of 0.1–0.5 M imidazole in buffer A. The *Pf*DXR was then applied to a cation-exchange column, 1mL HiTrap SP FF column (Cytiva), which had been equilibrated with buffer B (20 mM HEPES-NaOH pH 6.0 and 50 mM NaCl). To ensure binding of *Pf*DXR to the cation-exchange column, the pH of the buffer solution was set at 6.0, well below the *Pf*DXR’s isoelectric point of 7.7. Subsequently, the column was washed with 10 column volumes of buffer B. *Pf*DXR was eluted with a linear gradient of 0.3–1.0 M NaCl in buffer B. The presence of *Pf*DXR was confirmed by SDS-PAGE on fractions that exhibited a notable absorption at 280 nm. Thereafter, the fractions containing *Pf*DXR were pooled and mixed with 20 mM HEPES-NaOH pH 6.0 at a volume ratio of 1:4 and then concentrated to 6 mg/mL using a centrifugal concentrator (30 kDa MWCO) (AS ONE, Osaka, Japan).

### 4.2. Crystallization

To obtain ternary (*Pf*DXR–Mn^2+^–inhibitor) and quaternary (*Pf*DXR–Mn^2+^–NADPH–inhibitor) complexes, a protein solution (6 mg/mL *Pf*DXR, 20 mM Tris-HCl pH 8.0, and 0.2 M NaCl) was mixed with the inhibitor solution without or with NADPH (20 mM, Tris-HCl pH 8.0, 0.2 M NaCl, 4 mM DTT, 4 mM MnCl_2_, 4 mM inhibitor, and 0 or 4 mM NADPH) at a volume ratio of 1:1. The crystallization was conducted using the hanging drop method, whereby 1 μL of the ternary or the quaternary complex solution was mixed with an equal volume of reservoir solution (0.1 M Tris-HCl pH 8.0, 14–20% (*w*/*v*) PEG6000, and 0.2 M CaCl_2_) and incubated at 20 °C. The resulting drops were suspended over 200 μL of the reservoir solution in 48-well plates. The crystals reached a typical final size of 0.03 × 0.03 × 0.25 mm^3^ within a timeframe of 3–7 days.

### 4.3. X-Ray Data Collection

For data collection under cryogenic conditions, the crystals in a droplet were transferred directly to the harvesting solution [0.07 M Tris-HCl pH 8.0, 14% (*w*/*v*) PEG6000, 0.14 M CaCl_2_, and 30% (*v*/*v*) glycerol] for a period of 10 s. The crystals were mounted in nylon loops, flash-cooled in liquid nitrogen, and loaded into a Unipuck. The data were collected by the rotation method at −173 °C using a DECTRIS EIGER X 4M detector (DECTRIS, Barden, Switzerland) (BL1A at the Photon Factory), a DECTRIS PILATUS3 S 6M detector (BL5A at the Photon Factory), a DECTRIS PILATUS3 S 2M detector (NW12A at the Photon Factory), or DECTRIS EIGER X 16M detectors (BL44XU at the SPring-8 and BL17A at the Photon Factory). Laue group and unit-cell parameters were determined using the xia2 3.8.6/DIALS 3.8 software package [28]. The data collection statistics are summarized in Appendix A.

### 4.4. Structure Determination

The initial phase determination of the MAMK150–quaternary complex of *Pf*DXR was conducted using a difference Fourier method with the coordinate set of the recently obtained high-resolution quaternary complex structure of *Pf*DXR [24] (PDB ID: 9JOB) as the starting model. This was achieved through the rigid-body refinement of the REFMAC5 [29] program, which is part of the CCP4 Program Suite 8.0 [30]. Model building and refinement at a resolution of 1.53 Å were conducted with the COOT [31] and REFMAC5 [29] programs. The stereochemistry of the model was verified using the RAMPAGE [32] program. The refined MAMK150–quaternary complex was then employed as a template for the structure refinement of the other complexes. Although all inhibitors were synthesized as racemic compounds, high-resolution crystal structure analyses revealed that only the *S*-enantiomers were bound to the active site of *Pf*DXR (Appendix A). In each complex, no electron density was detected that would suggest the presence of *R*-enantiomers in the active site. This observation is consistent with our findings from previous generations of inhibitors [15,24]. The refinement statistics are summarized in Appendix A. The atomic coordinates for the reported structures have been deposited with the Protein Data Bank under the following accession codes: 9JZ7, 9JZ8, 9JZ9, 9JZA, 9JZB, 9JZC, 9JZD, and 9JZE (Appendix A).

For the MAMK89 complexes, the resolution cutoff criteria (*I*/σ(*I*) > 2.0 at the outer shell of the resolution range) was too stringent for the data reductions of the ternary and quaternary complexes of MAMK89 in our recent study [24]. Consequently, the data reduction was redone using the criteria of cc-half > 0.5 at the outer shell of the resolution range. As a result, the resolutions were enhanced from 1.60 Å and 1.44 Å to 1.50 Å and 1.39 Å, respectively, for the ternary and quaternary complexes. The crystal structures of the ternary and quaternary complexes were subjected to a further round of refinement against the higher-resolution intensity data. In consequence, the PDB IDs 8JNV and 8JNW, which pertain to the ternary and quaternary complexes, respectively, have been rendered obsolete and updated to 9JOA and 9JOB.

### 4.5. Computational Analyses

The crystal structures obtained in this study were preprocessed for the computational analyses using Protein Preparation Wizard [33] in the Maestro program of Schrödinger Suite 2024-1 (Schrödinger, LLC, New York, NY, USA, 2024). This process involved the addition of hydrogen atoms and optimization of the hydrogen-bonding network. All crystallographic water molecules were removed, and the protein structures were subjected to restrained minimization with an RMSD cutoff value of 0.3 Å using the OPLS4 force field.

Subsequently, the structures were subjected to molecular mechanics with generalized Born and surface area solvation (MM-GBSA) analysis in the Prime program [25,26] of Schrödinger Suite 2024-1. This approach is a rapid force field-based technique that employs a generalized Born and surface area continuum (implicit) solvation solvent model to determine the binding free energy (Δ*G*_bind_) from the difference between the energies of the complex, ligand, and protein in solution. In this instance, MM-GBSA was calculated with the default settings, which entail the utilization of an OPLS4 force field and a variable-dielectric generalized Born (VSGB) solvent model. All ligand atoms were designated as flexible during the calculations. The results of the MM-GBSA analyses are summarized in Table 1.

## 5. Conclusions

The aim of this study is to gain insight into the molecular mechanism underlying the structure–activity relationship of the reverse *N*-phenylalkyl derivatives of fosmidomycin. In order to achieve the objective, a series of MAMK89 analogs was employed, as outlined in Figure 1. These included alterations in the linker length (C2, C4, and C5) of the *N*-phenylalkyl group, as well as the introduction of substituents to the terminal phenyl moiety (*p*-methyl and *p*-methoxy) of the *N*-phenylpropyl group. The high-resolution crystallographic analyses were successfully performed for all complexes. The structural analyses in this study revealed that the nanomolar activities of the reverse *N*-phenylalkyl fosmidomycin derivatives against *Pf*DXR are dependent on diverse binding modes. In our previous study, the pronounced inhibitory activity and high binding affinity can be attributed to interactions between the phenyl moiety of the *N*-phenylpropyl group of MAMK89 and the subpocket in the active site of *Pf*DXR [24]. In the present study, this phenomenon can be attributed to the interactions of the tip of the *N*-substituent with the residues from the active site core and from the flexible loop. The reason for this discrepancy is that when the alkyl linker of the *N*-phenylalkyl group of the inhibitor was too short (MAMK150) or too long (MAMK251) or when a functional group was introduced onto the phenyl moiety (MAMK431 and MAMK433), the terminal phenyl moiety was unable to be accommodated in the subpocket. Despite the discrepancies in the mode of interaction, the cumulative effect of the interactions between the *N*-substituent and the active site residues is responsible for the pronounced inhibitory activities exhibited by the series of inhibitors.

## Figures and Tables

**Figure 1 molecules-30-00072-f001:**
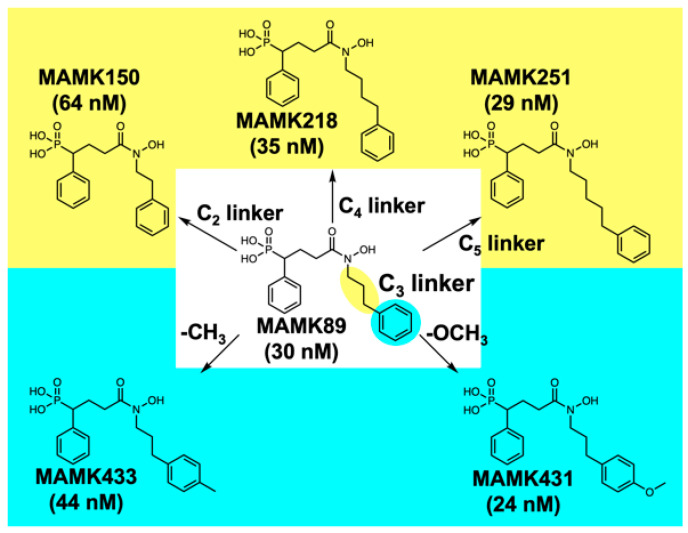
The inhibitors utilized in this study for crystallographic analyses in complex with *Pf*DXR. The IC_50_ values of the inhibitors against the enzymatic activity of *Pf*DXR from our recent study [24] are provided in parentheses. The MAMK89 analogs with the following structural modifications were chosen for the crystal structure analyses: (i) alterations in the linker length (yellow) and (ii) introduction of substituents to the terminal phenyl moiety (light blue) of the *N*-phenylpropyl group.

**Figure 2 molecules-30-00072-f002:**
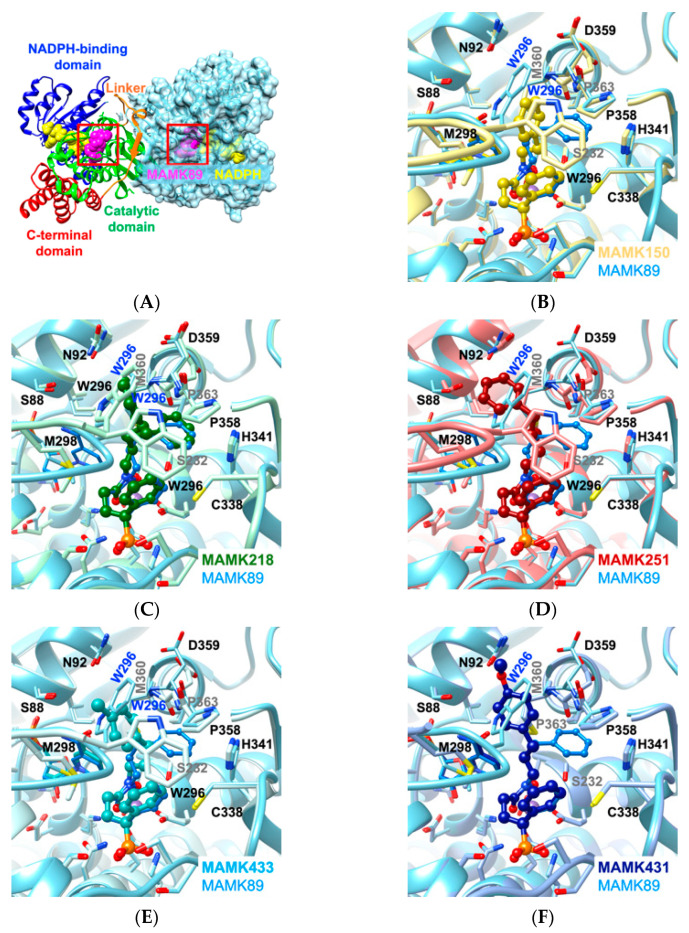
The crystal structures of *Pf*DXR in complex with the *N*-phenylalkyl fosmidomycin derivatives. (**A**) The overall structure of the MAMK89–quaternary complex of *Pf*DXR. One subunit is shown in a ribbon model, and the other in a surface model. The substrate/inhibitor binding site of each subunit is indicated by red squares. The bound MAMK89 and NADPH molecules are shown in space-filling models and colored magenta and yellow, respectively. (**B**) MAMK150–quaternary complex (yellow and pale yellow). (**C**) MAMK218–ternary complex (green and pale green). (**D**) MAMK251–ternary complex (red and pale red). (**E**) MAMK433–quaternary complex (light blue and pale light blue). (**F**) MAMK431–quaternary complex (dark blue and pale dark blue). In panels (**B**–**F**), the MAMK89–quaternary complex (sky blue and pale sky blue) is also shown for comparison.

**Figure 3 molecules-30-00072-f003:**
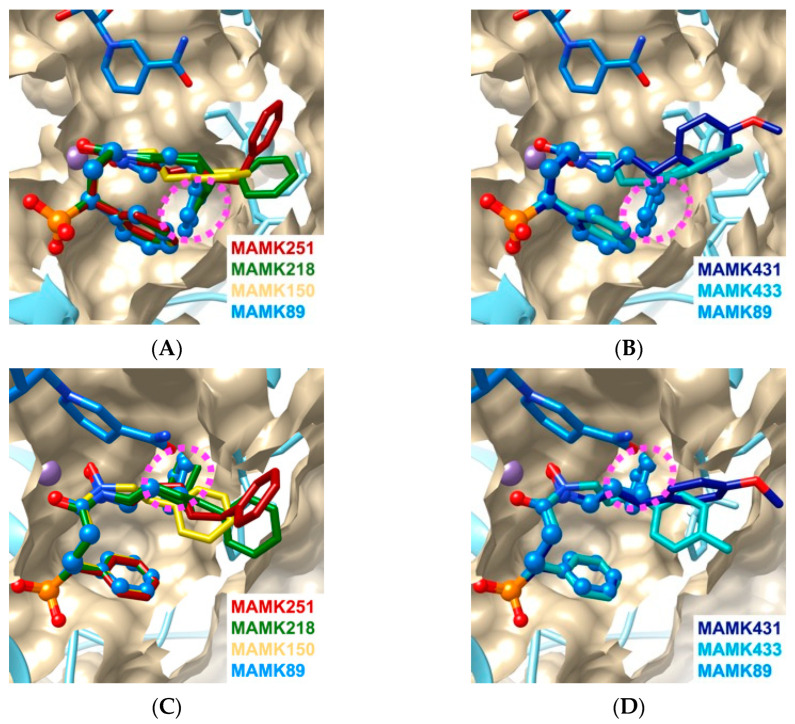
The superpositions of the binding modes of the *N*-phenylalkyl fosmidomycin derivatives in the active site of *Pf*DXR. Inhibitor molecules are colored in accordance with Figure 2. MAMK89 is depicted as a ball-and-stick model, while the remaining inhibitors are illustrated as stick models. (**A**) A comparison of linker length. The *N*-phenylalkyl groups of MAMK89 (sky blue) and MAMK218 (green) assume bent conformations and occupy the subpocket (indicated by a dashed ellipsoid, magenta), whereas MAMK150 (yellow), MAMK218 (green), and MAMK251 (red) adopt curved conformations. (**B**) A comparison of the phenyl moiety with/without *para*-substituent. The *N*-phenylpropyl group of MAMK89 (sky blue) assumes a bent conformation and is bound within the subpocket (indicated by a dashed ellipsoid, magenta), whereas MAMK433 (light blue) and MAMK431 (dark blue) adopt curved conformations. (**C**) View of the molecules in (**A**) from a direction rotated by 50° around the *x*-axis. (**D**) View of the molecules in (**B**) from a direction rotated by 50° around the *x*-axis.

**Figure 4 molecules-30-00072-f004:**
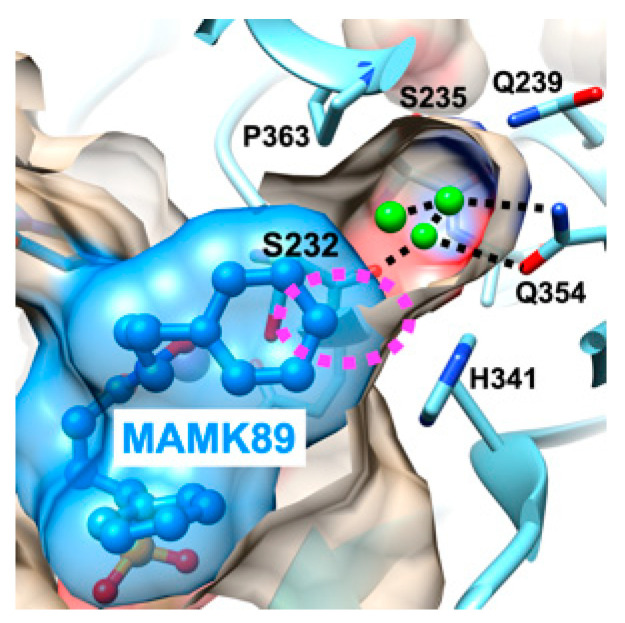
A surface representation of the subpocket in the active site of the MAMK89–quaternary complex of *Pf*DXR. The *para* position of the phenyl moiety of the *N*-phenylpropyl group of MAMK89 is shown in the center and indicated by a dashed ellipsoid depicted in magenta. The bound water molecules located at the rear of the subpocket are depicted in green, and potential hydrogen bonds surrounding these molecules are indicated by dashed lines.

**Figure 5 molecules-30-00072-f005:**
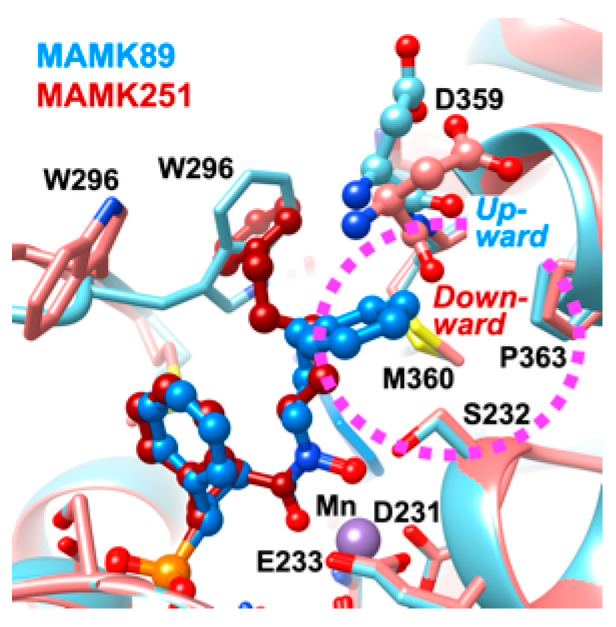
The conformational variation of the main-chain carbonyl group of Asp359 located at the entrance to the subpocket within the active site of *Pf*DXR. The MAMK89–quaternary and MAMK251–ternary complexes are illustrated and colored in accordance with the conventions in Figure 2. The subpocket is indicated by a dashed ellipsoid depicted in magenta. The carbonyl groups of Asp359 adopt upward and downward conformations, respectively, in the case of the MAMK89 and MAMK251 complexes.

**Table 1 molecules-30-00072-t001:** The binding free energy of each inhibitor to the active site of the quaternary complex of *Pf*DXR estimated by means of MM-GBSA analyses.

Inhibitor (PDB ID)	Disordered Residues	Conformation of the *N*-Substituent of the Inhibitor	IC_50_ (nM) [24]	MM-GBSA Δ*G*_bind_ (kcal/mol) Subunit A, Subunit B
Fosmidomycin (3AU9)	-	-	160	−14.70, −17.11
MAMK89 (9JOB)	-	Bent	30	−42.84, −41.59
MAMK150 (9JZ8)	-	Curved	64	−39.14, −37.11
MAMK218 * (9JZ9)	293-295	Bent	35	−39.66, −40.38
MAMK218 * (9JZ9)	293-295	Curved	35	−34.15, −34.41
MAMK251 (9JZC)	293-295	Curved	29	−38.45, −40.55
MAMK433 (9JZD)	293-295	Curved	44	−39.44, −37.54
MAMK218 ^#^ (9JZA)	293-297	Bent	35	−21.59, −19.50
MAMK218 ^#^ (9JZA)	293-297	Curved	35	−22.12, −18.47
MAMK431 ^$^ (9JZE)	293-296	Curved	24	−32.94, −29.58

* The values for the MAMK218–ternary complex (9JZ9) are presented here for reference, as Trp296 in the MAMK218–quaternary complex (9JZA) was observed to be disordered and the MM-GBSA score for the MAMK218–quaternary complex was not properly estimated (^#^). Similarly, Trp296 was observed to be disordered in the MAMK431–quaternary complex (9JZE). Consequently, the MM-GBSA score was not properly estimated for the MAMK431–quaternary complex as well (^$^).

## Data Availability

The coordinates and structure factor amplitudes for the crystal structures of the ternary and quaternary complexes of *Pf*DXR were deposited in the Protein Data Bank with the following accession codes: MAMK89–ternary, 9JOA; MAMK89–quaternary, 9JOB; MAMK150–ternary, 9JZ7; MAMK150–quaternary, 9JZ8; MAMK218–ternary, 9JZ9; MAMK218–quaternary, 9JZA; MAMK251–ternary, 9JZB; MAMK251–quaternary, 9JZC; MAMK433–quaternary, 9JZD; and MAMK431–quaternary, 9JZE.

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
