# Peer review of "The Diverse Binding Modes Explain the Nanomolar Levels of Inhibitory Activities Against 1-Deoxy-d-Xylulose 5-Phosphate Reductoisomerase from Plasmodium falciparum Exhibited by Reverse Hydroxamate Analogs of Fosmidomycin with Varying N-Substituents"

_molecules, 2024, doi:10.3390/molecules30010072_

Round 1
Reviewer 1 Report
Comments and Suggestions for Authors
I would like to congratulate the authors on their work. The results presented clearly show the potential of this type of study in understanding the interactions between molecules of biological interest.
Some comments that I believe are appropriate for readers to better understand the manuscript:
1- Page 4, line 170: how many water molecules were considered in the study?
2 - Same page; lines 178 and 179, and on page 5, lines 205 and 206: the authors should explain what they mean in the sentence: "...thus excluded from their respective atomic coordinates."
3 - The authors should explain the MM-GBSA methodology because not all readers are familiar with the methodology,
4 - On page 9, line 354, the authors should present evidence on the "reduction in degrees of freedom on the PfDXR molecule."
5- Was the pKa of the phosphate group considered in the hypothesis presented on page 11, line 432?
6 - The authors should explain the reason for using two pH values: 6.0 and 8.0.
7- On page 12, line 491, the authors report that only the S enantiomer binds to the active site of PfDXR. Should the authors comment on the R enantiomer?
8 - I found the conclusion of the manuscript too short. The authors have good results that they should present in this section of the manuscript.
Reviewer 2 Report
Comments and Suggestions for Authors
In this work, the authors study the binding properties of different analogs of fosmidomycin to 1-deoxy-D-xylulose 5-phosphate reductoisomerase from Plasmodium falciparum. The paper is well-written, and both the main and supplemental data are well-curated. Minor text adjustments are required.
Title: If possible, make the title more concise.
Keywords: IspC abbreviation is never explained.
Keywords: Is a comma missing between DXP and fosmidomycin?
Results: It's not clear in chapter 2.1 if the new original data are described or if it’s just a summary of the data from references [10, 14]. If the reported information was already published, please move it to the Introduction or Discussion. Another concern: citations [10] and [14] are both review papers, it would be better to refer to original research papers.
Results, line 150: In the phrase “..at distances of approximately 2.1 A,” is it better to indicate exact distances?
Table 1: For fosmidomycin, IC is 160 nM and delta-G are -14.7 and -17.11. For two MAMK218 bent and curved, IC is 35 nM and delta-G are very similar to fosmidomycin. Please discuss why, for such similar delta-G, the IC is so different.
Discussion, lines 292 and 357: The mention of Figure 1 is better shifted to the end of the phrase. As it is written now, “the objective of the study is Figure 1.”
Methods: All temperatures are reported in Kelvin. Consider using Celsius units.
Methods, line 454: “Thereafter, the fractions containing PfDXR were pooled …”. It is not clear how the presence of PfDXR was determined in those fractions. Which method is used to quantify the specific presence of PfDXR? Describe it in detail in the Methods.
Reviewer 3 Report
Comments and Suggestions for Authors
In this article, Takada et al. solved 8 crystal structures of PfDXR in complex with a series of fosmidomycin derivatives. Following up on a recent student where the authors found that phenylalkyl substituents on fosmidomycin increased its relative affinity for PfDXR, the authors hypothesized that a new hydrophobic pocket exists to accommodate this extra extension of PfDXR. Using fosmidomycin derivatives with small shifts in linker length and phenyl substituents, this new structural analysis tried to understand how this new hydrophobic pocket could be best used to increase fosmidomycin affinity. Although the final picture is more complex than initially thought, the authors provide a thorough, molecular explanation for how the movement and fit between multiple subsections of PfDXR and fosmidomycin derivatives combine to determine their relative affinity. The science presented in the article is strong and does not need further editing prior to publication. The presentation of the results could however use improvement to better accentuate the key components within the structure and how these shift with each fosmidomycin derivative.
Suggested improvements:
1. In the results section, the authors begin by summarizing each of the co-crystal structures individually and then illustrating these structures in Figure 2. In these paragraphs and structures, the authors list off a large number of residues and contacts but lose the main story about what differs between each of the various inhibitors.
a. The authors should add a table or figure that summarizes the side-chain interactions within each co-crystal structure and better illustrates how these shift between structures. Maybe a table with a list of all residues with any interactions and then colored markers for the type of interaction seen within that structure.
b. The structures in Figure 2 do not show or illustrate the key loop movements within the structures and the multiple binding modes that are central to explaining the relative affinity. This reviewer suggests that the authors redo Figure 2 and switch from a simple image of each bound structure to images that better illustrate changes and shifts within and between structures. These figures could be moved to the SI.
c. For at least one of the images from Figure 2, it would also be helpful to show the binding surface to illustrate globally how and where these inhibitors are located.
2. Figure 3B is meant to illustrate the bent and curved conformations of the multiple inhibitors. However, in this single view, it is difficult to see how the inhibitor rotates between these two conformations. The authors should consider showing multiple angles (rotated ~90) to better see the rotations.
